# Knowledge, Attitudes and Practice Behaviour of Midwives Concerning Periodontal Health of Pregnant Patients

**DOI:** 10.3390/ijerph17072246

**Published:** 2020-03-27

**Authors:** Jennifer Gia-linh Nguyen, Shanika Nanayakkara, Alexander C. L. Holden

**Affiliations:** 1Faculty of Dentistry, University of British Columbia, Vancouver, BC V6T 1Z3, Canada; 2Faculty of Medicine and Health, University of Sydney School of Dentistry, Westmead, NSW 2145, Australia; shanika.nanayakkara@sydney.edu.au (S.N.); alexander.holden@sydney.edu.au (A.C.L.H.)

**Keywords:** midwifery, oral health, pregnancy, periodontal disease, adverse pregnancy outcomes, prenatal care, maternal health, interprofessional education, provider practices, women’s health

## Abstract

Maternal gum disease is associated with adverse pregnancy outcomes such as preterm birth and low birthweight. This study aims to evaluate the knowledge, attitudes and practice behaviour of Australian midwives regarding the periodontal health of pregnant women to inform interprofessional antenatal care. This was an observational, cross-sectional study. We circulated an online questionnaire to Australian midwives from August 2018 to February 2019. Key outcome variables were knowledge, attitudes, and practice behaviours related to oral health. Key predictor variables were years of practice, practice location, and dental history. We summarized responses with frequency tables and assigned tallied scores for analysis using non-parametric statistical tests. 100 responses were analysed, including from rural (n = 23) and urban (n = 77) midwives. Eighty percent of midwives agreed that maternal dental care can positively affect pregnancy outcomes. Fluoridated toothpaste use (19.1%) was incorrectly answered to prevent gum disease more often than psychological stress control (7.9%), a correct answer. Rural midwives demonstrated a significantly higher knowledge score (p = 0.001) and significantly more positive practice behaviours towards oral health (p = 0.014) than urban midwives. Australian midwives have positive attitudes towards antenatal oral health but misunderstand gum disease aetiology and prevention. This study highlights areas to improve interprofessional education for optimal oral health and pregnancy outcomes.

## 1. Introduction

Oral infections, especially periodontal disease, are associated with an increased level of systemic inflammation and may contribute to poorer outcomes for systemic diseases such as diabetes, respiratory disease, cardiovascular disease, and osteoporosis [1,2,3,4,5,6]. Periodontal disease is a chronic, inflammatory bacterial infection of the tissues surrounding the teeth, which progresses from gingivitis—initial inflammation of the gums [4,7]. Women are at an increased risk of developing oral diseases such as periodontal disease and dental caries during pregnancy, due to elevated concentrations of oestrogen and progesterone, increased systemic levels of inflammatory cytokines, nausea and gastroesophageal reflux disease, and increased incidence of snacking [8,9,10,11,12]. Up to 75% of women develop gingivitis during pregnancy, while 20%–50% of women have periodontal disease which worsens during pregnancy [13,14]. Periodontal disease during pregnancy is correlated with adverse pregnancy outcomes such as preterm delivery, low birthweight, and pre-eclampsia [11,15,16,17,18,19]. Periodontal disease is a direct consequence of untreated gingivitis, and is treatable and preventable with early intervention, oral health education and screening [7,20,21]. In this light, monitoring and maintaining periodontal health in pregnant patients presents an opportunity for interprofessional collaboration in health care to improve patient outcomes [20,22,23,24].

Previous research has investigated the attitudes and practice behaviours of health professionals, including obstetricians, gynaecologists, nurses, midwives and dentists, towards oral health during pregnancy. Although obstetricians and gynaecologists were aware of the association between periodontal disease and adverse pregnancy outcomes, their attitudes and practice behaviour regarding antenatal oral health did not match accordingly [25,26]. Most obstetricians and gynaecologists accepted that there is minimal discussion of oral health during care of pregnant females, and they identified limited time for consultations as a major reason for this approach [14,27,28]. Other studies reported that, although most midwives and nurses referred pregnant patients for dental care, many midwives and nurses were not aware of appropriate referral pathways to a dentist and felt inadequately trained to provide oral health information to pregnant patients [29,30,31]. Most health professionals believed it is mainly the dentist’s role to provide oral health information to pregnant patients [14,30,32,33].

It has been reported that, although dentists are aware of the association between poor oral health and advanced pregnancy outcomes, pregnant women are often not referred to the dentist for routine dental care, and misconceptions also persist regarding the safety of dental care during pregnancy [26,34,35]. Furthermore, while finances are another barrier for pregnant women to seeking routine dental care, most of them are also unaware of the importance of oral health during pregnancy and assumed that poor oral health was a normal change during pregnancy [29,36,37,38,39]. Overall, this paints a picture of unclear knowledge amongst health care professionals regarding antenatal oral care management and presents areas to improve interprofessional coordination of care and referral for antenatal oral health. Improved inter-disciplinary communication and frameworks for collaboration can lead to better patient outcomes in antenatal oral health [40,41,42].

Midwives, as antenatal and post-natal care providers, tend to care for pregnant patients over a longer period of time than obstetricians, providing pregnant patients with continuity of care, health education, and opportunities for interprofessional collaboration for improved patient outcomes [43,44,45]. In this light, midwives can play an important role in providing antenatal oral health screening and education, as well as dental referrals [40,46]. Furthermore, recent surveys have also reported that Australian women are positive towards receiving oral health advice from midwives during pregnancy [38,47].

International guidelines in the United States, the UK and Europe encourage antenatal oral health screening, education and referral for dental care starting from early pregnancy, as there is an association between antenatal periodontal disease and adverse pregnancy outcomes [23,48,49,50]. Australian national antenatal care guidelines were updated in 2012 to promote oral health screening for all pregnant patients, as oral health is important for maternal health, and dental treatment can be provided safely during pregnancy [51,52]. Despite guideline changes to include oral health in midwifery training, Australian midwifery programs contain minimal formal content regarding oral health [40,53,54,55]. Previous research found that midwives believe oral health is important during pregnancy but have inadequate knowledge regarding antenatal oral health care and education [30,31].

The purpose of this study is to evaluate the current knowledge, attitudes, and practice behaviours of midwives in Australia regarding the periodontal health of pregnant women to inform and identify areas to improve antenatal care.

## 2. Materials and Methods

All subjects gave their informed consent for inclusion before they participated in the study. The study was conducted in accordance with the Declaration of Helsinki, and ethical clearance for this study was granted by the University of Sydney Human Research Ethics Committee (Project No. 2017/659).

A three-part questionnaire with 31 items was developed based on an amalgamation of previously published, validated questionnaires [14,28,30] (Appendix A). The questionnaire was then sent to an expert panel, consisting of a periodontist, an oral health therapist, and a midwife currently practicing in Australia, for survey pretesting and feedback prior to circulation to participants [56]. The first part of the questionnaire was comprised of eight multiple-choice questions regarding knowledge of clinical signs and pathogenesis of periodontal disease, as well as its link to pregnancy. The second part was comprised of twelve questions in total—eight Likert-scale questions on participants’ attitudes and opinions towards periodontal health, and four close-ended questions on participants’ practice behaviour related to oral health during pregnancy. The third section, with ten close-ended questions and one Likert-style question, evaluated participants’ socio-demographics and oral health status and care. Further questionnaire modifications were made based on the expert panel’s suggestions, to highlight the role of dental plaque debridement in ameliorating periodontal disease progression, as opposed to the role of dental plaque control in preventing dental caries. This is because the difference between dental caries and periodontal disease is often misunderstood by the general public.

Question phrasing and structure were revised in this study to minimize response bias and optimize response rate, based on the survey’s online method of distribution [56,57,58]. Questions were structured to only cover one idea or topic per item to provide clarity of interpretation and more accurate response data. Survey questions were also worded to avoid using double negatives, avoid biased phrasing that could encourage participants to give certain answers, and avoid questions about hypothetical scenarios or future intentions, as these question formats tend to provide unreliable data about respondent behaviour [57,58]. Likert-style questions with a five-point scale were used to give a more accurate representation of both participants’ general attitude and the intensity of their opinions on statements in the survey [57] (p. 770). A close-ended response format was used to provide more reliable uniformity of responses for analysis and ease of administration for an online survey format. Since the study questionnaire covered sensitive topics such as attitudes and clinical practice behaviours, participants were more likely to give accurate and truthful answers with a close-ended format [57,58]. This study used an online self-completion method of distribution for the survey to reduce response bias from having an interviewer present, while allowing participants to answer sensitive questions more honestly [57] (p. 775).

Facebook groups were used to sample participants due to their cost effectiveness, wide audience reach, and increased timeliness in gathering responses through this medium [56]. Using an online method to gather participants also encourages accessibility for midwives located across Australia, including in rural areas. The alternative method of contacting midwives via hospital lists and private practices would have involved greater administrative complexity via multiple interstate Australian bodies to gain ethics approval, more time, and a projected smaller participant reach. Study participants were recruited from six Facebook groups for professional midwives in Australia, entitled “Midwifery Group Practice in Australia”, “Registered Nurses & Midwives in Australia”, “Call the Student Midwife (Australia)”, “Midwifery Contracts in Australia”, “Midwifery Students and Novice Midwives of Melbourne, Australia”, and “South Australian Midwifery Students”. These Facebook groups were selected for the study as they were tailored towards midwives in Australia, had the largest group membership, and were active groups. Participants could circulate the questionnaire to their colleagues to participate in the study, even if the colleague was not a member of one of the listed midwifery Facebook groups. The eligibility criteria for participation were either being a midwife currently practicing in Australia or being a current midwifery student at an Australian institution. The questionnaire was circulated by posting a link to the online survey in the midwifery Facebook group. Members of the Facebook groups were reminded one month and two months after initial circulation of the questionnaire to participate in the study. Study data was collected from August 2018 to February 2019. Questionnaires with answers to all the questions were included in the analysis.

Questionnaire data was collected on REDCap, an encrypted, secure server hosted by the University of Sydney, and participant responses were de-identified through this process [59,60]. Geographical data was mapped using QGIS. Key outcome variables are knowledge, attitudes, and practice behaviour related to oral health. Key predictor variables are practice setting (urban versus rural), years of professional experience, and personal dental history.

Statistical analyses were performed using IBM SPSS Statistics (version 23, IBM SPSS Inc., Chicago, IL). Frequency tables were used to summarize the responses. Questionnaire responses regarding knowledge (Part 1) and attitudes and practice behaviours (Part 2) were assigned tallied scores for further analysis using non-parametric statistical tests. A higher knowledge score indicated better knowledge and a higher attitude score indicated more positive attitudes toward oral health. For all statistical analysis, the significance level was set at p ≤ 0.05.

## 3. Results

One hundred and four complete questionnaires were collected, and one hundred questionnaires were included for analysis. Four questionnaires were eliminated because the participants were not currently working midwives in Australia. The mean age of participants was 37.5 years old ± 12.34 (minimum = 20, maximum = 62). The median number of years of professional experience was less than 10 years (Table 1). Geographical distribution of study participants across Australia was diverse, with respondents from northern, eastern, western, and southern regions of Australia, including Tasmania (Figure A1). More participants were from areas with higher population concentrations, such as Sydney and Melbourne. In terms of practice setting, most participants (77.0%) practiced in an urban setting, while 23.0% participants practiced rurally. The majority of participants (85.0%) practiced in a hospital setting, while some (13.0%) practiced at a mixture of hospital and private practice. In terms of dental history, most participants (87.0%) visited the dentist between every 6–18 months, and most participants rated their oral health as excellent (22.0%) or good (58.0%). Only 23.0% of participants had been diagnosed with periodontal disease in the past, and 22.0% had received treatment for periodontal disease.

### 3.1. Knowledge and Attitudes Towards Periodontal Disease

Sixty-three percent of participants correctly understood that periodontal disease is an inflammatory condition involving bacterial infection. However, nearly a third of participants (31.8%) incorrectly believed that periodontal disease is a degenerative process. Participants correctly listed gingival bleeding (28.9%), tooth mobility (21.6%) and tooth loss (21.0%) as clinical signs of periodontal disease, but caries (16.0%) was often also mistaken as a sign of periodontal disease. In terms of aetiological factors for gum disease, poor oral hygiene (23.5%) and smoking (20.0%; 18.2%) were often correctly identified as aetiological factors associated with initiation and progression of gum disease. However, tooth decay (10.0%) was often also incorrectly chosen as an aetiological factor, more often than genetics (9.0%) and pregnancy (8.6%).

The majority of participants (83.0%) believed that periodontal disease influences pregnancy outcomes. Furthermore, most participants (98.0%) agreed that periodontal diseases can be prevented or arrested during pregnancy. In terms of prevention of periodontal disease, the correct options of effective toothbrushing technique (25.4%), interdental brush use (25.1%), and smoking cessation (22.5%) were chosen, as well as the incorrect answer of fluoridated toothpaste use (19.1%), with similar frequency (Table 2). Interestingly, the correct answer of psychological stress control was chosen only 7.9% of the time as part of periodontal disease prevention.

Most respondents (81.0%) agreed that periodontal disease can have an adverse effect on pregnancy outcomes, and 80.0% of participants agreed that treatment of periodontal disease during pregnancy positively affects pregnancy outcomes (Table 3).

In terms of oral health assessment during a midwifery visit, most respondents (69.0%) believed that asking pregnant patients about their oral health is within the routine practices of a midwife. Sixty percent of participants routinely ask questions related to oral health during consultation with pregnant patients. Most participants (79.0%) also provide oral-health-related information during a consultation with pregnant patients routinely or if the patient is considered at risk. However, only 16.0% of participants felt that they were up to date on the topic of oral health and pregnancy. Nearly half of the participants (49.0%) routinely refer patients to their dentist for check-ups. Most participants (81.0%) believed that conducting an examination of the oral cavity during pregnancy is outside the routine practices of a midwife.

Regarding the statement “There is insufficient time to address oral health during a care visit with a midwife”, 42.0% of participants agreed, while 40.0% disagreed.

Knowledge and attitude scores were not significantly correlated with midwives’ age, number of years of professional practice, nor frequency of dental visits (p>0.05). There was a weak correlation between knowledge and attitude scores which was not statistically significant (r = 0.147, p = 0.136).

### 3.2. Attitudes and Practice Behaviour

A higher attitude score significantly correlated with positive practice behaviour regarding antenatal oral health, such as providing oral-health-related information (r = 0.403, p < 0.01) and routinely asking oral-health-related questions (r = 0.458, p < 0.01). Specific attitudes that were significantly correlated with midwives providing oral-health-related information were that “Treating periodontal disease can positively affect pregnancy outcomes” (p = 0.012), “An oral examination is within routine midwifery practice” (p = 0.018), “There is sufficient time to address oral health during an appointment” (p < 0.001), “Asking about oral health is within routine midwifery practice” (p < 0.001), and “Routine dental care is important during pregnancy” (p = 0.012). Attitudes that were significantly correlated with midwives referring patients for dental care included the beliefs that “Asking about oral health is within routine midwifery practice” (p = 0.027) and “Routine dental care is important during pregnancy” (p = 0.002).

### 3.3. Midwives and Practice Location

Knowledge and attitude scores correlated with practice location. Rural midwives had a significantly higher knowledge score (r = 0.313, p = 0.001) and a higher attitude score (r = 0.017, p = 0.865) than urban midwives. However, more urban midwives felt up to date on the topic of oral health and pregnancy (*X^2^* = 16.56, df = 4, *p* = 0.002) than rural midwives. More rural midwives felt that conducting an oral examination was within the routine scope of a midwife (*X^2^* = 16.08, df = 4, *p* = 0.003), compared to urban midwives. There was no significant difference between urban and rural midwives regarding having insufficient time to address oral health during appointments. In terms of practice behaviour, significantly more rural midwives asked oral-health-related questions (*X^2^* = 5.413, df = 1, *p* = 0.02), conducted an oral examination for pregnant patients (*X^2^* = 9.884, df = 1, *p* = 0.002), and provided oral-health-related information (*X^2^* = 8.56, df = 2, *p* = 0.014) than urban midwives.

## 4. Discussion

One of the objectives of the national law that governs the structure and regulation of many of the health professions in Australia is to ensure a dynamic workforce that is responsive to the needs of Australian society [61]. The role of midwives within the interdisciplinary healthcare team is one that is important to oral health; many patients who may not seek care from dental professionals will access care from midwives. It is therefore a missed opportunity if midwives are not suitably equipped with the knowledge and skills to deliver basic oral health promotion activities, provide oral health education, and act as advocates for oral healthcare and the access of services. By investigating midwives’ knowledge, attitudes, and practice behaviour concerning the periodontal health of pregnant patients, this study represents the first step in identifying the gaps in current oral disease knowledge and building a stronger interdisciplinary health care team for improved patient outcomes.

All participants in this study were female and the average age was 43 years old. This is similar to the national demographics of Australian midwives, where average age was 44 years old and 99% of practicing midwives were female [62]. Most participants in this study trained as a midwife during or after the Australian antenatal care guidelines were updated to include oral health—44.0% of participants practiced for less than 10 years as a midwife, while 32.0% of participants were current midwifery students [51].

Midwives in this study understood basic clinical signs of periodontal disease, but were unclear about the definition and aetiology of periodontal disease and the difference between periodontal disease and dental caries—two distinct oral diseases. For example, participants understood that periodontal disease is an inflammatory bacterial infection (63.0%), but also incorrectly defined periodontal disease as a degenerative process (31.8%). In addition, midwives correctly identified key clinical signs of periodontal disease such as gingival bleeding (28.9%) and tooth mobility (21.6%), but often misidentified dental caries (16.0%) as a sign of periodontal disease. Midwives also incorrectly identified excessive sugar consumption (14.7%) and tooth decay (10.0%)—both associated with dental caries—as aetiological factors for periodontal disease initiation more often than pregnancy (8.6%) or genetics (9.0%). While dental caries and periodontal disease are both important diseases that should be addressed during pregnancy, the aetiology and therefore the treatment methods for these two oral diseases are quite different. In terms of periodontal disease prevention, use of fluoridated toothpaste (19.1%) was often incorrectly answered. Fluoridated toothpaste use is a preventative factor for dental caries, not periodontal disease. Control of psychological stress (7.9%) was also rarely answered as a preventative measure for periodontal disease, although this is a factor midwives can help address during appointments. This underscores some misunderstanding amongst midwives regarding basic preventative measures and systemic inflammatory effects of periodontal disease. Similar results, including incorrectly believing periodontal disease development is associated with tooth decay and excessive sugar consumption, were also found in previous studies of maternal care providers [27,28,30]. This highlights gaps in basic oral health knowledge amongst Australian midwives, despite national regulation changes to include oral health in midwifery training [51].

Although Australian midwives agreed (91.1%) that periodontal diseases influence pregnancy outcomes, there was also a lack of awareness regarding the consequences of antenatal periodontal disease on pregnancy outcomes. The most common responses for adverse pregnancy outcomes associated with periodontal disease were pre-term birth (34.9%), low birthweight (20.3%), and spontaneous abortion (17.7%)—these were low percentages, considering all possible answers were correct and participants were allowed to choose more than one response.

This study found no correlation found between years of professional practice and knowledge, attitudes, and practice behaviour regarding antenatal oral health in spite of aforementioned national regulation changes [51]. Furthermore, the majority of participants (76%) in this study had practiced for 10 years or less, and therefore should have been trained with the updated midwifery curricula. This suggests that Australian midwifery curricula have not been adequately adjusted to improve midwives’ oral health knowledge. A previous study of Australian midwives in New South Wales found an association between oral health knowledge and years of experience, but only when midwives practiced for more than 20 years, suggesting this was likely not due to midwifery curricular changes mandated by Australian national guidelines [31].

Midwives’ knowledge of antenatal oral health was only weakly correlated with their overall attitudes and practice behaviour (p>0.05). Most respondents (81.0%) agreed that periodontal disease can have an adverse effect on pregnancy outcomes and most (80.0%) agreed that treatment of periodontal disease during pregnancy positively affects pregnancy outcomes. This is similar to previous surveys on midwives regarding the importance of oral health during pregnancy [29,30,31,53]. However, only 39.0% of midwives routinely gave oral-health-related information to pregnant patients and less than half (49.0%) routinely referred pregnant patients for dental care.

### 4.1. Attitudes and Practice Behaviour

Overall, positive attitudes amongst midwives towards oral health were strongly correlated with positive practice behaviour regarding antenatal oral health (p < 0.01). In particular, the beliefs that “Asking about oral health is within routine midwifery practice” and that “Routine dental care is important during pregnancy” were strongly correlated with providing oral-health-related information during a consultation (p < 0.001; p = 0.012, respectively) and referring pregnant women to the dentist (p = 0.027; p = 0.002, respectively).

Only 24.2% of midwives in this study believed asking pregnant patients about their oral health is beyond the routine practices of a midwife. However, only 15.5% of midwives felt up to date on the topic of oral health and pregnancy. Similar views regarding antenatal oral health were found in surveys of Australian midwives practicing in urban Sydney and Victoria [31,63]. Surveys of midwives in Sydney, Australia and the USA also found approximately only 20% of midwives felt up to date regarding maternal oral health [30,31]. 

21.0% of midwives in this study never referred pregnant patients for a dental consult. Possible barriers to midwives addressing oral health during a routine visit include unclear referral pathways for dental care, the midwife’s knowledge of oral health, and perceived lack of time during the appointment [29,30]. Previous research found that a major barrier to midwives addressing oral health during consultations was a lack of confidence to answer patient questions related to oral health [29]. Midwives in this study were also divided about whether there is time to address oral health during routine pregnancy consultation with patients. A total of 42.7% believed there is no time, while 39.8% believed there is time to address oral health. There was a strong positive correlation between midwives feeling they had sufficient time to address oral health and providing oral-health-related information (p < 0.001). Surveys on midwives in Melbourne, Australia found that having oral health information pamphlets available would reduce the amount of consultation time needed to address oral health, make midwives more likely to address oral health, and increased likelihood of patient referral to a dentist [29]. George et al. also found a positive correlation between midwives having information brochures regarding oral health and discussing the importance of oral health with pregnant patients [31].

### 4.2. Differences in Practice Location

Interestingly, rural midwives had a significantly better knowledge score (p = 0.001) regarding periodontal disease and pregnancy, and more positive attitudes and practice behaviours (p = 0.014) regarding antenatal oral health than urban midwives. However, significantly more urban midwives felt they were up to date on the topic of antenatal oral health compared to rural midwives (p = 0.002). Significantly more rural midwives asked oral-health-related questions (p = 0.02), conducted an oral examination (p = 0.002), and provided oral-health-related information (p = 0.014) than urban midwives. Factors contributing to this disparity in knowledge, attitudes, and practice behaviour should be investigated in future research. Contributing factors may include educational training and opportunities available, patient case complexity, and work culture attitudes regarding interdisciplinary care. 

### 4.3. Limitations

A weakness of this study was possible sample selection bias, as this was an online survey using social media to recruit participants. Participants were also not compensated and answered the survey during their spare time. Thus, study participants may be midwives who use social media more often and have more positive attitudes towards oral health. In this light, the results of this study may be more applicable towards midwives within this demographic. This study also did not include a question regarding participants’ prior oral health knowledge from sources other than midwifery training, such as the pilot Midwifery Initiated Oral Health (MIOH) program [53]. However, the MIOH program is still in the early pilot stage and has trained a limited number of midwives in NSW and Victoria, therefore this was considered a negligible effect for this study [53].

### 4.4. Future Directions

Australian pilot initiatives have aimed to implement oral health curricula for midwives and midwifery students in Melbourne and Sydney [53]. However, recent studies found that Australian midwives still do not feel they have adequate knowledge about antenatal oral health and appropriate pathways to refer pregnant patients for dental care [29,31,46]. It remains yet to be seen whether these oral health education programs can be implemented nationally, and whether they will improve Australian midwives’ knowledge of antenatal oral health and modify practice behaviours. Recent studies found that a major barrier to oral health care for pregnant patients was access to dental services and referral by midwives to a dentist [26,29,30]. Lim et al. found that although professional development courses regarding oral health were helpful for midwives to increase their knowledge and confidence, midwives forgot most information regarding oral health a few weeks after taking the course [29]. This implies that a clearer, more structured pathway of interprofessional communication, standardized information dissemination, and referral is needed between midwives and dentists regarding care of pregnant patients. Research has found information brochures regarding antenatal oral health were helpful for midwives to increase likelihood of addressing oral health [29,31].

The 2018 Australian Government Department of Health guidelines encourage inter-disciplinary collaboration for patient-centred care to improve patient outcomes [52]. Puertas et al. recommend that as part of general initial care, pregnant women should be informed that periodontal disease may increase their risk for adverse pregnancy outcomes [21]. They also recommend a brief oral health history to be gathered and a simple oral examination for pregnant women as part of initial general screening. Gaining awareness of midwives’ perceptions regarding oral health is important for strengthening patient health care in a multidisciplinary teamwork setting. Management of antenatal oral diseases must move away from an isolated, singular approach tackled solely by dental professionals to a patient-centred interprofessional team approach for improved patient outcomes [41,42,64].

Future investigations of this topic may involve using a larger participant pool and including a survey question about participants’ pre-existing knowledge and learning source regarding oral health. Further research may also investigate factors influencing the difference in knowledge and practice behaviour between urban and rural midwives, such as training opportunities available and work culture regarding interdisciplinary care. It would be helpful to assess whether the Australian guideline changes and consequent modifications to midwifery training lead to any improvement in patient-centred interprofessional health care and midwives’ understanding of antenatal oral health. 

## 5. Conclusions

Australian midwives have positive attitudes towards the importance of maternal oral health during pregnancy and believe that dental care can positively impact pregnancy outcomes. Viewing oral health as within routine midwifery practice was significantly associated with providing oral-health-related information and referring patients for dental care. Significantly more rural midwives provided oral-health-related information, asked oral-health-related questions, and conducted an oral examination for pregnant patients than urban midwives. Rural midwives also had significantly better knowledge of maternal gum disease than urban midwives. Overall, there are areas for improvement in midwives’ understanding of the aetiology and prevention for gum disease during pregnancy. An improved understanding of maternal gum disease can give midwives a clearer understanding of how they can aid in improving pregnant women’s oral health. The findings of this study show that further development of the Australian midwifery curricula is needed to support midwives in developing basic competencies to understand the fundamental signs and causes of oral disease, provide basic oral health information, and refer patients for further dental care. Midwives can play a key role in educating pregnant patients about oral health and connecting these patients at risk of gum disease with access to needed dental care. Interprofessional collaboration is critical for achieving optimal oral health and improving pregnancy outcomes.

## Figures and Tables

**Table 1 ijerph-17-02246-t001:** Demographic characteristics and dental history of participants.

		%	n
Gender	Female	100.0%	100
	Male	0.0%	0
	Other	0.0%	0
Number of years in midwifery practice	Current midwifery student	32.0%	32
	Less than 10 years	44.0%	44
	10–20 years	11.0%	11
	21–30 years	10.0%	10
	31–40 years	6.0%	6
	40+ years	0.0%	0
Current occupation	Currently practicing midwife in Australia	68.0%	68
	Current midwifery student	32.0%	32
	Unemployed	0.0%	0
	Retired midwife	0.0%	0
Midwifery practice location	Urban	77.0%	77
	Rural	23.0%	23
Midwifery practice setting	Hospital	85.0%	85
	Hospital and private practice	13.0%	13
	Private practice	2.0%	2
How would you rate your oral health?	Excellent	22.0%	22
	Good	58.0%	58
	Neutral	14.0%	14
	Poor	6.0%	6
	Very poor	0.0%	0
How often do you visit the dentist?	Every 6 months	33.0%	33
	Every 12 months	33.0%	33
	Every 12-18 months	21.0%	21
	Only if in pain	12.0%	12
	Never	1.0%	1
Have you ever been diagnosed with periodontal disease?	Yes	23.0%	23
	No	77.0%	77
Have you ever been diagnosed and received treatment for periodontal disease?	Yes	22.0%	22
	No	78.0%	78

**Table 2 ijerph-17-02246-t002:** Midwives’ knowledge of periodontal disease ^1^.

Survey Question	Respondent Answer	%	n
Definition of periodontal disease (N = 154)	Inflammation and bacterial infection	63.0%	97
	Degenerative process	31.8%	49
	Auto-immune disorder	2.0%	3
	Osteoporosis	2.0%	3
	Metastatic process	1.3%	2
Clinical signs associated with periodontal disease (N = 343)	Gingival bleeding	28.9%	99
	Tooth mobility	21.6%	74
	Tooth loss	21.0%	72
	Caries	16.0%	55
	Alveolar bone destruction	12.5%	43
Risk factors for gum disease initiation (N = 409)	Poor oral hygiene	23.5%	96
	Smoking	20.0%	82
	Excessive sugar consumption	14.7%	60
	Tooth decay	10.0%	41
	Dental plaque	14.2%	58
	Genetics	9.0%	37
	Pregnancy	8.6%	35
Risk factors for gum disease progression (N = 396)	Poor oral hygiene	23.5%	93
	Smoking	18.2%	72
	Excessive sugar consumption	16.4%	65
	Tooth decay	11.6%	46
	Dental plaque	15.9%	63
	Genetics	3.0%	12
	Pregnancy	11.4%	45
Oral signs often related to pregnancy (N = 162)	Gingival bleeding	61.1%	99
	Gingival overgrowth	20.4%	33
	Caries	9.3%	15
	Tooth loss	9.3%	15
Do periodontal diseases influence pregnancy outcomes? (N = 192)	Yes – Increased incidence of preterm birth	34.9%	67
	Yes – Increased incidence of low-weight newborn	20.3%	39
	Yes – Increased incidence of spontaneous abortion	17.7%	34
	Yes – Increased incidence of low genital-tract infection	10.4%	20
	Yes – increased incidence of pre-eclampsia	7.8%	15
	No	8.9%	17
Are periodontal diseases preventable during pregnancy? (N = 100)	Yes – They can be prevented or arrested during pregnancy	98.0%	98
	No – They’re an expected side effect during pregnancy	2.0%	2
Periodontal diseases can be prevented by: (N = 382)	Effective toothbrushing technique	25.4%	97
	Using dental floss or interdental brushes	25.1%	96
	Smoking cessation	22.5%	86
	Using fluoridated toothpaste	19.1%	73
	Control of psychological stress	7.9%	30

^1^ Respondents were allowed to choose more than one answer for these questions, thus the number of responses per question varied depending on the total number of responses chosen per question.

**Table 3 ijerph-17-02246-t003:** Midwives’ attitudes and practice behaviours regarding maternal periodontal disease.

Likert-Scale Questions	Strongly Agree %(N)	Agree % (N)	Neutral % (N)	Disagree % (N)	Strongly Disagree % (N)
For patients with periodontal disease, periodontal treatment is beneficial for improving oral health	82.0%(82)	18.0%(18)	0%(0)	0%(0)	0%(0)
Periodontal disease can have an adverse effect on pregnancy outcomes	41.0%(41)	40.0%(40)	11.0%(11)	8.0%(8)	0%(0)
Treatment of periodontal disease during pregnancy positively affects pregnancy outcomes	38.0%(38)	42.0%(42)	17.0%(17)	2.0%(2)	1.0%(1)
Asking pregnant patients about their oral health is outside the routine practices of a midwife	9.0%(9)	16.0%(16)	6.0%(6)	34.0%(34)	35.0%(35)
Conducting an examination of the oral cavity during pregnancy is outside the routine practices of a midwife	41.0%(41)	40.0%(40)	5.0%(5)	10.0%(10)	4.0%(4)
It is important for a pregnant woman to receive routine dental care during her pregnancy	57.0%(57)	37.0%(37)	5.0%(5)	1.0%(1)	0%(0)
There is not sufficient time to address oral health during a care visit with a midwife	16.0%(16)	26.0%(26)	18.0%(18)	33.0%(33)	7.0%(7)
I am up to date on the topic of oral health and pregnancy	3.0%(3)	13.0%(13)	28.0%(28)	42.0%(42)	14.0%(14)
**Additional questions**		**% (N)**			
I routinely ask questions related to oral health during consultation with pregnant patients	Yes	60.0%(60)			
	No	40.0%(40)			
I routinely perform a visual oral examination during consultation with pregnant patients	Yes	7.0%(7)			
	No	93.0%(93)			
I provide oral-health-related information during consultation with pregnant patients	Routinely	39.0%(39)			
	If patient is considered at risk	40.0%(40)			
	Never	21.0%(21)			
I refer patients to their dentist for a check-up	Routinely	49.0%(49)			
	If patient is considered at risk	33.0%(33)			
	Never	18.0%(18)			

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
