# Peer review of "Knowledge, Attitudes and Practice Behaviour of Midwives Concerning Periodontal Health of Pregnant Patients"

_ijerph, 2020, doi:10.3390/ijerph17072246_

Round 1

Reviewer 1 Report

This paper is of public health interest and value. 

The major questions that I have is around the methods employed, the how informed consent was obtained, the validity of the questionnaire and the sampling. 

 1. Please check hyphen _ in line 86

2. How did the authors arrive at establishing the validity of this questionnaire for these participants in this context.  

3. How was anonymity of the participants maintained if at all in this process.

4. A major need for clarification is the sampling of participants.  How many midwives are there in Australia. What are the sources of data of practicing and registered midwives.  How and why were Facebook sites and these facebook sites chosen over any other sources? this also needs to be raised in the limitations of the study.

5. it is not clear why there is a call for a rotation of students in rural areas. Is there an educational rationale and evidence for this. is it a realistic recommendation? 

6. Perhaps consider including the sampling and representativity of this sample in the limitations.

7. The article may give the impression unwittingly that only periodontal disease and not dental caries need to be addressed especially during pregnancy. Perhaps the writing around this can be more nuanced.  

6. The recommendations are generalised and could be more specific in engaging at an educational and health level with issues dealing with midwives having the competencies and skills to examine the mouth, make a preliminary oral diagnosis, recommend treatment and referral to the oral health care team.

Reviewer 2 Report

This is a nicely written paper about a nice clearly designed study. I don't think birthweight should be hyphenated. You speak about potential associations in literature review and appropriate as these associations are disputed but then your questionnaire asks as if it is fact.
